# No Time to Die: How Cytomegaloviruses Suppress Apoptosis, Necroptosis, and Pyroptosis

**DOI:** 10.3390/v16081272

**Published:** 2024-08-09

**Authors:** Yingqi Deng, Ana Águeda-Pinto, Wolfram Brune

**Affiliations:** Leibniz Institute of Virology (LIV), 20251 Hamburg, Germany; dengyingqi92@126.com (Y.D.); ana.pinto@leibniz-liv.de (A.Á.-P.)

**Keywords:** cytomegalovirus, cell death, apoptosis, necroptosis, pyroptosis, immune evasion

## Abstract

Viruses are obligate intracellular pathogens as their replication depends on the metabolism of the host cell. The induction of cellular suicide, known as programmed cell death (PCD), has the potential to hinder viral replication and act as a first line of defense against viral pathogens. Apoptosis, necroptosis, and pyroptosis are three important PCD modalities. Different signaling pathways are involved in their execution, and they also differ in their ability to cause inflammation. Cytomegaloviruses (CMV), beta-herpesviruses with large double-stranded DNA genomes, encode a great variety of immune evasion genes, including several cell death suppressors. While CMV inhibitors of apoptosis and necroptosis have been known and studied for years, the first pyroptosis inhibitor has been identified and characterized only recently. Here, we describe how human and murine CMV interfere with apoptosis, necroptosis, and pyroptosis signaling pathways. We also discuss the importance of the different PCD forms and their viral inhibitors for the containment of viral replication and spread in vivo.

## 1. Cytomegalovirus Infection and Pathogenesis

Cytomegaloviruses (CMVs) are large, enveloped, double-stranded DNA viruses that infect mammals, including humans, in a species-specific manner. All CMVs are related members of the *Betaherpesvirinae* that have diverged as they coevolved with their hosts. They have adapted to subvert the immune system of their specific host, leading to widespread distribution and long-term coexistence [1,2,3].

Human cytomegalovirus (HCMV) is an opportunistic pathogen responsible for significant morbidity and mortality in immunocompromised patients such as hematopoietic stem cells and solid-organ transplant recipients [4]. HCMV infection can also be transmitted vertically from the mother to the unborn child, where it can cause long-term neurological sequelae. In fact, congenital HCMV infection is the most frequent infectious cause of birth defects worldwide. Due to the species-specificity of the CMVs, HCMV infection and pathogenesis cannot be studied in animals [5]. Therefore, CMVs of mice, rats, guinea pigs, and rhesus macaques are used to study CMV pathogenesis in its natural host organism. Although the rhesus macaque CMV and the guinea pig CMV models are especially well suited to study congenital HCMV infection, murine CMV (MCMV) has emerged as an excellent animal model where the interplay of viral modulators and host defense pathways can be investigated by using genetic mutants. Therefore, MCMV infection has been used to elucidate numerous mechanisms of infection, pathogenesis, and immune response to CMV in vivo.

The CMV genomes are 200–240 kbp linear double-stranded DNA molecules that encode more than 200 proteins. Most of them are involved in the modulation of virus–host interaction and the subversion of innate and adaptive immunity [6,7]. HCMV exhibits a broad tropism, infecting several cell types, with one of its pivotal features being its ability to induce different infection outcomes depending on the infected cell. Lytic infection occurs mainly in epithelial, endothelial, fibroblast, and smooth muscle cells and results in the release of infectious viral particles and spread to the neighboring cells [8]. For HCMV, hematopoietic cells of the myeloid lineage are a major site of latency. Reactivation from latent reservoirs can lead to recurrent disease, particularly in immunocompromised individuals [4]. After primary infection, HCMV induces innate immune responses involving the release of pro-inflammatory cytokines and interferons (IFNs) and activation of natural killer (NK) cells. CMV infection also triggers strong humoral and cell-mediated adaptive immune responses [9]. Like all herpesviruses, CMVs remain latent within their respective hosts for life, being controlled but not cleared by the antiviral immune response.

Innate immunity acts as the host’s first line of defense against CMV infection, activating different stress signals and defensive pathways. In particular, the activation of programmed cell death pathways provides a quick response against pathogens and results in the death and removal of infected cells. Because CMVs are highly dependent on the viability of infected cells to complete their viral life cycle, these viruses have evolved various mechanisms to suppress cell death starting from the early period after infection [10]. Here, we provide an overview of the molecular events triggered by CMV that lead to the induction of apoptosis, necroptosis, or pyroptosis and describe the evasion strategies used by these viruses to avoid or delay these responses. We also review the biological importance of viral cell death inhibition for infection and pathogenesis in the mouse model.

## 2. Overview of Programmed Cell Death Pathways Activated during CMV Infection

The evolution of CMVs with their natural hosts is a complex interplay that has occurred over millions of years, where the CMVs adapted to subvert the immune system and establish persistent infections without causing severe symptoms in immunocompetent hosts. This homeostasis has been shaped by host–pathogen interactions that include host recognition of the virus infection, while the virus evolved to evade the host’s immune responses.

Cell death can be traditionally classified into accidental cell death (ACD) or programmed cell death (PCD). Tightly regulated intracellular signaling pathways and their effector molecules orchestrate PCD, while ACD arises from unforeseen cellular injury. The best-studied forms of PCD are apoptosis, necroptosis, and pyroptosis, and these present very different features (Table 1) [11]. Apoptotic cell death has long been considered a critical cell death process for host defense by eliminating infected cells and preventing the spread of intracellular pathogens within the host [12]. Both physiological and pathogen-derived stimuli can induce apoptosis, and two main pathways result in its activation: the extrinsic and intrinsic pathways. In general, apoptosis leads to cell shrinkage, chromatin condensation, and cell fragmentation [12]. Apoptotic cells are rapidly cleared by phagocytic cells in tissues resulting in an inflammatory-silent form of cell death [13]. In the last two decades, other PCD forms such as necroptosis and pyroptosis have been identified as alternative death pathways that dominate under specific conditions [14,15]. Morphologically and mechanistically different from apoptosis, these PCDs are characterized by membrane rupture and initiation of a strong inflammatory response [11,16]. Although these pathways are often described as distinct signaling cascades, an increasing number of studies have revealed a substantial level of cross-talk between PCD forms leading to alternative outcomes, for instance, when caspase inhibition converts extrinsic apoptosis to necroptosis [17,18]. A complex form of PCD, termed PANoptosis, may be triggered via Z-nucleic acid binding protein 1 (ZBP1) when viruses fail to suppress cell death signaling (reviewed in [19,20]).

### 2.1. Inhibition of Apoptosis by CMVs

Apoptosis is the best-described form of programmed cell death, and its multifaceted functions contribute to the host’s ability to fight viral infections and maintain homeostasis. The extrinsic apoptosis pathway begins with the activation of cell-surface death receptors of the tumor necrosis factor (TNF) receptor family by their specific extracellular death ligands (Figure 1). These death receptors include TNF receptor 1 (TNFR1, activated by TNFα), FAS (activated by Fas ligand, FasL), and TNF-related apoptosis-inducing ligand (TRAIL) receptors 1 and 2 (activated by TRAIL) [21,22]. Activated FAS and TRAIL receptors recruit their adaptor protein Fas-associated death domain protein (FADD), which binds to apoptotic pro-caspase-8 or -10, and the cellular FADD-like IL-1β-converting enzyme (FLICE) inhibitor protein (cFLIP), forming the death-inducing signaling complex (DISC) [23,24,25,26]. Long and short cFLIP isoforms control the activation of the caspase cascade, promoting apoptotic cell death [27]. In contrast to Fas and TRAIL receptors, TNFR1 recruits the TNFR-associated protein with a death domain (TRADD) as an adaptor protein, which subsequently recruits FADD, TNF-associated factor-2 (TRAF2), the receptor-interacting protein kinase 1 (RIPK1) and the cellular inhibitor of apoptosis (cIAPs) [28,29,30]. After this complex is formed, FADD binds and activates caspase-8, which in turn activates the effector caspases-3/6/7 resulting in apoptotic cell death [31,32,33].

Intrinsic apoptosis (also known as the mitochondrial apoptosis pathway) is activated when intracellular stress interferes with the balance of the B-cell lymphoma 2 (BCL-2) family of pro- and anti-apoptotic proteins (Figure 1). The small “BH3-only” proteins are activated in response to stress or, in the case of BID (BCL-2 homology domain 3 (BH3)-interacting domain death agonist), connect the extrinsic with the intrinsic pathway [34,35]. They bind to anti-apoptotic proteins such as BCL-2, BCL-XL, and myeloid cell leukemia 1 (MCL-1), which control the activity of the pro-apoptotic BCL-2-associated X protein (BAX) and BCL-2 antagonist/killer (BAK) [36,37,38,39]. When cellular stress is sensed by cells, the anti-apoptotic proteins are inhibited, resulting in the release of BAX and BAK. These proteins then oligomerize and cause mitochondrial outer membrane permeabilization (MOMP) and the release of cytochrome c (cyt*C*), SMAC/Diablo, and HtrA2/Omi [40,41,42,43]. Mitochondrial release of SMAC/Diablo and HtrA2/Omi overcomes the activity of inhibitor of apoptosis proteins (IAPs). In the cytosol, cyt*C* binds to apoptotic protease-activating factor 1 (Apaf-1), forming an oligomeric complex [44,45]. This complex, also known as apoptosome, promotes autocatalytic activation of caspase-9, which subsequently activates the same effector caspases as caspase-8 in the extrinsic apoptotic pathway [46].

Over the years, the mechanisms of CMV inhibition of apoptosis have been studied in detail (Figure 1). In the course of CMV infection, multiple cellular processes can promote the release of pro-apoptotic stimuli. For instance, large amounts of viral glycoproteins are synthesized during viral infection. Unfolded or misfolded proteins in the secretory pathway can activate ER stress and initiate an unfolded protein response (UPR) [47]. If not resolved by counter-regulatory mechanisms, the UPR leads to the induction of apoptosis (in that case termed “terminal UPR”). During HCMV infection, the viral protein UL38 plays a key role in inhibiting apoptosis induced by ER stress (Figure 1). UL38 induces the accumulation of activating transcription factor 4 (ATF4) and suppresses the activation of c-Jun N-terminal kinase (JNK), a protein known to inhibit the action of anti-apoptotic proteins like BCL-2 and BIM [48,49,50]. Perturbation of mitochondrial metabolism by viral infection can also result in increased levels of oxidative stress and apoptosis. During the early stages of infection, the β2.7 non-coding RNA is produced by HCMV in large amounts, helping it counteract oxidative stress and maintain the mitochondrial membrane potential [51]. Because this RNA can be produced in high amounts with no need for translation, it has been proposed that this strategy allows HCMV to target mitochondria function quickly throughout the course of infection [51].

The CMVs are also known to encode proteins that directly interfere with cell surface expression of death receptors. The MCMV m166 viral gene product specifically inhibits TRAIL expression and is critical for replication in vivo [52]. Because HCMV glycoprotein UL141 is also capable of binding to TRAIL death receptors [53], restriction of the TRAIL/TRAIL-DR pathway seems to play a key role in the CMV life cycle. Direct control of death receptor signaling also contributes to effective viral pathogenesis. CMVs encode the viral inhibitor of caspase-8 activation (vICA), which directly binds to caspase-8, inhibiting its activity (Figure 1). The HCMV vICA is encoded by open reading frame (ORF) UL36, and its homolog in MCMV is encoded by ORF M36 (Figure 1). These proteins directly interact with the death domain of pro-caspase-8 to block caspase-8 activation, thereby reducing apoptotic cell death [54,55]. Caspase-8 suppression by M36 proteins was shown to be necessary for successful infection of monocyte-derived cells and control of viral dissemination [56]. Interestingly, a mutant MCMV expressing UL36 or overexpressing a dominant-negative FADD instead of M36 fully rescued the replication defect of the MCMV M36 deletion mutant in macrophages, showing that this form of apoptosis inhibition is conserved among the CMVs [57,58]. HCMV Immediate Early 2 protein (IE2) can also interfere with apoptosis signaling by inducing the expression of c-FLIP in human retinal pigment epithelial cells, an apoptotic inhibitor downstream of the FasL/FAS cascade [59,60].

MOMP is a central event in the intrinsic apoptotic pathway, and CMVs are known to encode viral inhibitors of pro-apoptotic BCL-2 family proteins. The viral mitochondria-localized inhibitor (vMIA), encoded by HCMV ORF UL37 exon 1 (UL37x1), binds and sequesters both BAX and BAK, thereby preventing MOMP and apoptosis [61,62,63,64] (Figure 1). In human fibroblasts, UL37x1 also inhibits mitochondrial serine protease HtrA2/Omi-dependent cell death, which is independent of apoptosis [65]. Although UL37x1 has an important anti-apoptotic role, it has also been shown to inhibit antiviral signaling in a step downstream of mitochondrial antiviral-signaling protein (MAVS) [66]. MCMV, by contrast, encodes two separate proteins to inhibit BAX and BAK. MCMV ORF m38.5, localized to the mitochondria, encodes a protein similar to HCMV vMIA that specifically inhibits BAX [64,67,68,69]. Later, a second inhibitor, encoded by ORF m41.1, was shown to associate with BAK at the mitochondrial membrane, acting as a viral inhibitor of BAK oligomerization (vIBO) [70] (Figure 1). Similarly, guinea pig CMV encodes specific inhibitors for BAX and BAK [71,72]. Different cells infected with MCMV mutants lacking m38.5 or m41.1 were shown to be sensitive to apoptosis induction [67,69,70], and MCMV replication in leucocytes was reduced when one of these gene products was absent [73,74]. In vivo, replication of an m41.1-deficient MCMV was attenuated in the liver and lungs of infected mice [73,74,75]. However, the same phenotype was not observed in mice infected with MCMV deleted of m38.5 [69]. Although single mutant studies disrupting m38.5 or m41.1 have shown that these are important for MCMV replication, the combined effect of both proteins for replication and pathogenesis was studied only recently. Combined suppression of BAX and BAK activation by m38.5 and m41.1 was shown to prevent intrinsic PCD, limit mitochondrial HtrA2/Omi signaling, and impair innate inflammation, which benefit sustained virus replication [76]. In vivo, MCMV dissemination to or replication to high titers in the salivary glands was compromised in mice infected with an MCMV mutant lacking both m38.5 and m41.1 [76].

### 2.2. Inhibition of Necroptosis by CMVs

Necroptosis can be induced by the activation of death receptors such as FAS and TNFR1 or by stimulation of the pattern recognition receptors Toll-like receptor 3 (TLR3) and TLR4 or the intracellular sensor ZBP1, also known as DAI or DLM1 [77]. In contrast to apoptosis, necroptosis leads to the release of intracellular contents and inflammation and occurs without caspase activation (Figure 2).

While TNFR signaling triggers the extrinsic apoptotic pathway, it can also induce necroptosis when the apoptotic initiator caspase-8 is inhibited [78,79]. When caspase-8 is inhibited, RIPK1 recruits RIPK3 through an RIP homotypic-interacting motif (RHIM)-dependent interaction, leading to the formation of a RIPK1–RIPK3 complex that initiates necroptosis [80,81]. Therefore, viral inhibition of caspase-8 by vICA blocks the extrinsic apoptosis pathway but sensitizes host cells to necroptosis. The activated RIPK3 subsequently recruits and phosphorylates the pore-forming protein mixed lineage kinase domain-like pseudokinase (MLKL) [82,83,84]. Phosphorylated MLKL oligomerizes and translocates to the plasma membrane, where it forms membrane-disrupting pores that lead to leakage of cytosolic content and inflammation [84,85]. Proteins that contain RHIMs play central roles in necroptotic cell death. Two other RHIM-containing proteins, ZBP1 and the TIR-domain-containing adaptor inducing interferon-β (TRIF), can activate RIPK3-dependent necroptosis. ZBP1 directly binds and activates RIPK3 through its RHIM, while the adaptor TRIF is recruited after stimulation of TLR3, leading to its interaction with RIPK3 through its RHIM domain [86,87,88] (Figure 2).

The M45 protein of MCMV is a potent cell death suppressor that contains a RHIM motif within its N-terminal domain and an induced protein aggregation motif (IPAM) located close to the C-terminus [89,90,91,92]. Although M45 strongly binds to RIPK1 with its C-terminal domain even when the RHIM-containing N-terminus is absent [93,94], the M45 RHIM domain is crucial for the inhibition of the necroptotic signaling. It inhibits RHIM-dependent activation of RIPK3, preventing necroptosis initiated by FAS or TNFR. Hence, it was named viral inhibitor of RIP activation (vIRA) [89,93] (Figure 2). As expected, MCMV expressing M45 with a mutated RHIM failed to inhibit necroptosis [95]. By interfering with RHIM–RHIM interactions, M45 can also inhibit ZBP1 and TRIF-mediated RIPK3 activation and subsequent necroptosis [87,88]. In MCMV-infected cells, Z-RNA resulting from newly transcribed viral RNA is sensed by ZBP1, which initiates necroptosis if not inhibited by M45 [96]. Besides RIPK1, M45 also binds strongly to the NF-κB essential modulator (NEMO), the regulatory subunit of the inhibitor of κB kinase (IKK) complex [93,94,97]. Both RIPK1 and NEMO are involved in the activation of the pro-inflammatory transcription factor NF-κB. The interaction of M45 with these proteins leads to the formation of insoluble protein aggregates. Aggregation of these proteins requires the IPAM located in the C-terminus of M45. It mediates self-interaction and is required for the formation of insoluble protein aggregates, which are targeted to autophagosomes for degradation in lysosomes [91]. Thus, M45 functions as a powerful inhibitor of both necroptosis and NF-κB activation. This ability to inhibit necroptosis is essential not only for MCMV infection of specific necroptosis-sensitive cell types but also for MCMV dissemination in vivo [95,98,99].

Similar to M45, the human alphaherpesviruses HSV-1 and HSV-2 R1 proteins (ICP6 and ICP10, respectively) also contain a RHIM domain and inhibit RIPK1/RIPK3-dependent necroptosis [100,101]. The HCMV protein homologous to M45, UL45, does not possess a RHIM domain, and an HCMV lacking UL45 does not exhibit the same phenotype as an M45-deficient MCMV. Although UL45 can inhibit RIPK1-mediated NF-κB activation by cooperating with the viral deubiquitinase UL48 [102], it does not impede necroptosis. In fact, the mechanism of necroptosis inhibition employed by HCMV is distinct from that of MCMV and targets a downstream step of RHIM protein signaling [103]. A recent study has shown that HCMV protein UL36 can function as a dual inhibitor of PCD: UL36 inhibits the extrinsic apoptosis pathway and blocks necroptosis by targeting MLKL and inducing its degradation [104] (Figure 2). Interestingly, UL36 was shown to bind to both human and murine MLKL. However, although UL36 was capable of reducing the levels of murine MLKL, it failed to inhibit necroptosis in murine cells [105].

### 2.3. Inhibition of Pyroptosis by CMVs

Pyroptosis is a lytic and highly inflammatory form of PCD, morphologically distinguishable from apoptosis. In pyroptosis, cells lose plasma membrane integrity, leading to cell swelling and rupture rather than the membrane blebbing in apoptosis. The detection of pathogen-associated molecular patterns (PAMPs) or damage-associated molecular patterns (DAMPs) by cellular sensors such as NOD-like receptors (NLRs), AIM2-like receptors (ALRs) and RIG-I-like receptors (RLRs) initiate assembly of the inflammasome. While various inflammasomes serve to combat a wide array of pathogens, NLRP3 inflammasome, AIM2 inflammasome, and RIG-I inflammasome have been identified as crucial mediators of host responses to viral infection (reviewed in [106,107,108]). After sensing DAMPs or PAMPs, receptors recruit the adaptor protein ASC (apoptosis-associated speck-like protein containing a CARD), correlating with the oligomerization of ASC into macromolecular aggregates to form ASC specks [109,110]. The oligomeric inflammasome complex leads to the autoproteolytic cleavage of pro-caspase-1 into active caspase-1 consisting of the cleavage products p20 and p10 [111]. Activated caspase-1 processes the pro-inflammatory cytokine precursors pro-IL-1β and pro-IL-18 into their mature forms [112,113,114]. Moreover, caspase-1 also cleaves Gasdermin D (GSDMD), a key pyroptotic pore-forming protein, into an N-terminal (GSDMD-N) and C-terminal domain (GSDMD-C) [112,113]. GSDMD-N directly interacts with membrane phospholipids, leading to conformational changes that trigger oligomerization and the formation of a ring-shaped transmembrane pore [115,116,117] (Figure 3). GSDMD shares a common pore-forming mechanism with the entire GSDM protein family [115,118]. The formation of transmembrane pores by cleaved GSDMD and further plasma membrane rupture executed by NINJ1 protein [119,120,121] result in pyroptotic cell death and are essential for the release of the mature cytokines and other cellular components in the extracellular environment [114]. The release of mature cytokines IL-1β and IL-18 and proteins or “alarmins” derived from damaged cellular organelles such as the high-mobility group box 1 protein (HMGB1) contribute greatly to host responses against infected pathogens [122,123].

While it has been known for a long time that CMVs encode potent apoptosis and necroptosis inhibitors, a CMV inhibitor of pyroptosis has only recently been identified. NLRP3 detects a broad range of microbial motifs, endogenous danger signals and environmental irritants. The DNA sensors AIM2 and IFI16 are able to detect viral DNA upon viral infection. During infection with different herpesviruses, these receptors are known to be important sensors of viral infection and inducers of inflammation [107]. MCMV-infected AIM2-deficient mice were shown to have reduced IL-18 serum levels, and AIM2-deficient THP-1 cells were unable to efficiently control HCMV infection [124,125]. Interestingly, Horan et al. have shown that during CMV infection in myeloid cells, the viral DNA can be detected in the cytosol by IFI16 after ubiquitination and degradation of the viral capsid [126]. Only recently, the MCMV protein M84 was identified as an inhibitor of AIM2 inflammasome activation and signaling. The M84 protein interacts with both AIM2 and ASC through their pyrin domain to inhibit ASC speck formation (Figure 3). In the same way, M84 can also interact with other pyrin-domain-containing proteins, such as IFI203 and IFI204 [127]. The AIM2 inflammasome is mainly expressed in cells of the myeloid lineage such as macrophages and dendritic cells [128]. Compared with the WT virus, an M84*stop* mutant displayed a clear growth defect in macrophages. However, in fibroblasts, which lack AIM2 inflammasome expression, the M84*stop* mutant replicated to the same titers as the WT virus. The growth defect of the M84*stop* mutant was rescued when the AIM2 inflammasome was blocked (i.e., in AIM2 and ASC-deficient immortalized bone marrow-derived macrophages (iBMDM) or the presence of a caspase-1 inhibitor), suggesting that the AIM2 inflammasome restricts MCMV replication in macrophages and that M84 counteracts this restriction [127]. MCMV infection initially leads to transient caspase-1 activation and pro-inflammatory cytokine release [124]. Later, M84 inhibits caspase-1 activation and the release of IL-1β and IL-18 from MCMV-infected macrophages. Additionally, M84 also inhibits the cleavage of GSDMD, thereby reducing pyroptotic cell death. M84-mediated AIM2 inflammasome inhibition is important for MCMV replication and dissemination to the spleen and liver of infected mice and also inhibits the release of the inflammasome-activated cytokine IL-18 in the early stage of MCMV infection [127]. The stimulation of caspase-1-dependent canonical and caspase-11-dependent noncanonical pyroptotic pathways was shown in MCMV-induced retinitis of mice with murine acquired immunodeficiency syndrome (MAIDS) [129]. However, whether M84 inhibits caspase-11-dependent noncanonical pyroptosis remains unknown and requires further investigation.

Whether HCMV similarly interferes with inflammasome signaling remains unknown. The HCMV proteins UL83 and UL84 are most similar to MCMV M84 [130]. It was previously shown that HCMV protein UL83 interacts with the IFI16 pyrin domain to inhibit IFI16-mediated DNA sensing [131]. A UL83-deficient HCMV induced higher levels of caspase-1 activation compared to WT HCMV. A subsequent study showed that UL83 interacts with AIM2 and reduces the level of mature IL-1β in transfection assays [132]. Whether or not UL83 inhibits pyroptosis has not been investigated. The HCMV IE2 protein is also involved in the inhibition of the pro-IL-1β transcription and, independently, the induction of IL-1β protein degradation [133]. Further work will be required to fully understand the roles of HCMV proteins in modulating inflammasome signaling and pyroptosis.

## 3. Concluding Remarks and Perspectives

CMVs have evolved an array of mechanisms to inhibit PCD pathways in order to promote their replication. CMV inhibitors of the three major PCD pathways are described in this review. In some cases, HCMV and MCMV encode homologous viral proteins that function mechanistically in a very similar way. HCMV encodes multiple factors to subvert apoptotic response and mitochondrial cell death including UL36 (caspase-8 inhibitor); UL37x1 (binding and sequestering of BAX/BAK), UL38, which protects infected cells from endoplasmic reticulum stress; and β2.7 gene RNA, which is expressed much earlier during infection. Regarding the inhibition of necroptosis, HCMV is capable of inhibiting MLKL, an inhibitory mechanism that is substantially different from the one employed by MCMV.

The infection of mice with MCMV is a highly useful and practicable model to study CMV infection and pathogenesis in vivo [134]. Several studies have investigated the biological importance of MCMV cell death suppressors in this model. They revealed that viral inhibition of necroptosis is of great significance as M45-mutant MCMVs are highly attenuated and barely replicate and disseminate in vivo [92,95,98]. A strong attenuation was also observed in the absence of M36, the viral inhibitor of extrinsic apoptosis [58,135,136]. By contrast, the attenuation of MCMV mutants lacking inhibitors of intrinsic apoptosis (i.e., m38.5 or m41.1) or pyroptosis (M84) is more moderate [69,73,74,75,127]. However, this should not be interpreted to mean that these cell death pathways and their inhibition are of minor importance. MCMV and other viruses would not have acquired and maintained such inhibitors during their evolution if it were not to their benefit. A recent study has compared the impact of the different apoptosis and necroptosis inhibitors in MCMV-infected macrophages [76]. The results of the study suggested that MCMV employs sequential, non-redundant suppression strategies to specifically modulate PCD responses. Sequential activation of M45, M36 and m38.5/m41.1 initially suppresses RIPK3-dependent necroptosis, followed by inhibition of caspase-8-dependent extrinsic apoptosis and, lastly, intrinsic apoptosis regulated by BCL-2 family proteins and their viral inhibitors [76].

Until recently, very little was known about the importance of pyroptosis inhibition during CMV infection. Pyroptosis is usually mediated by inflammatory caspases and associated with the release of mature IL-1β and IL-18 [137]. MCMV M84 inhibits pyroptosis by inhibiting AIM2 inflammasome assembly and caspase-1-mediated cleavage of IL-1β, IL-18, and GSDMD [127]. It seems likely that HCMV encodes a similar inhibitor, but this remains to be demonstrated. As all of the N-terminal domains of GSDM family proteins can induce pyroptosis, sometimes without caspase activation [115], the term ‘pyroptosis’ has been re-defined as a GSDM-dependent type of cell death [138,139]. Considering the less-characterized functions of other Gasdermin proteins and their mechanisms of activation in different cell types during CMV infection, additional viral pyroptosis inhibitors might be identified in the future.

A recently described cytoplasmic multimeric protein complex named PANoptosome triggers a combined form of PCD dubbed PANoptosis [140]. The Kanneganti laboratory showed that ZBP1 interacts with RIPK3, caspase-6, and the NLRP3 inflammasome to form the ZBP1 PANoptosome complex that drives PANoptosis [141]. Later, the same laboratory defined another PANoptosome including AIM2, pyrin, and ZBP1 [142]. While the concept of PANoptosis is intriguing, many questions remain to be addressed. For example, it is unknown whether PANoptosomes are formed during CMV infection, whether the known PCD inhibitors are sufficient to inhibit all aspects of PANoptosis, or whether additional mechanisms of viral interference exist.

Overall, the interplay between PCD and CMV infection is complex and dynamic. Understanding these interactions is essential for developing strategies to combat CMV infections and designing novel antiviral therapies.

## Figures and Tables

**Figure 1 viruses-16-01272-f001:**
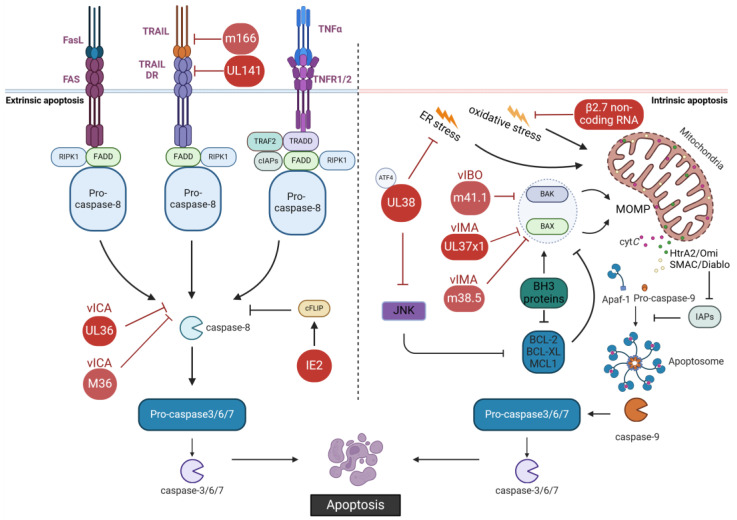
Inhibition of apoptosis by CMVs. The MCMV m166 protein inhibits TRAIL expression and HCMV UL141 prevents the expression of TRAIL death receptors on the cell surface. HCMV IE2 interferes with apoptosis signaling by inducing expression of cFLIP, an apoptotic inhibitor. HCMV protein UL36 and its homologous protein M36 in MCMV inhibit extrinsic apoptosis as viral inhibitors of Caspase-8 activation (vICA). HCMV protein UL37x1 and MCMV protein m38.5 are viral mitochondria-localized inhibitors of apoptosis (vMIA), which interfere with BAX. The m41.1 protein of MCMV inhibits apoptosis as a viral inhibitor of BAK oligomerization (vIBO). The HCMV β2.7 non-coding RNA counteracts oxidative stress and maintains the mitochondrial membrane potential. The HCMV protein UL38 inhibits apoptosis by inducing ATF4 accumulation to resolve ER stress. Additionally, UL38 curbs the activation of JNK, which suppresses anti-apoptotic BCL-2 and activates pro-apoptotic BH3 proteins. Created with BioRender.com.

**Figure 2 viruses-16-01272-f002:**
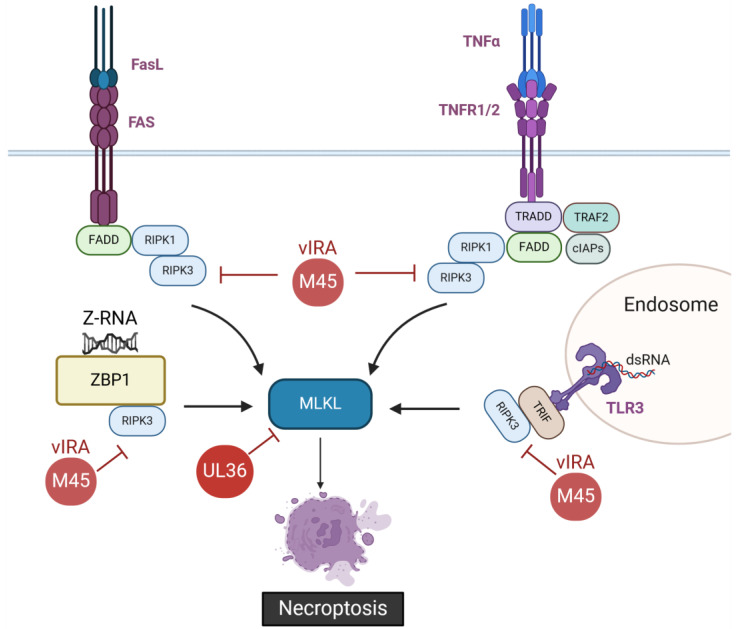
Inhibition of necroptosis by CMVs. MCMV protein M45, the viral inhibitor of RIP activation (vIRA), binds and inhibits RIPK3 activation through its RIP homotypic interaction motif (RHIM). M45 thereby inhibits RIPK3-MLKL-mediated necroptosis. HCMV protein UL36 inhibits necroptosis by targeting MLKL and inducing its degradation. Created with BioRender.com.

**Figure 3 viruses-16-01272-f003:**
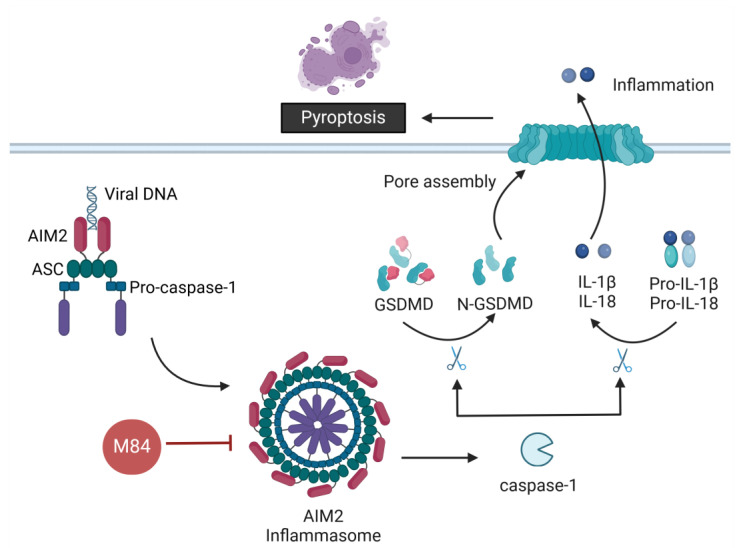
Inhibition of AIM2 inflammasome-mediated pyroptosis by MCMV. The M84 protein of MCMV interacts with AIM2 and ASC to inhibit inflammasome complex assembly. M84 thereby prevents downstream caspase-1 activation, the release of IL-1β and IL-18, GSDMD cleavage, and pyroptotic cell death. Created with BioRender.com.

**Table 1 viruses-16-01272-t001:** The main features of apoptosis, necroptosis, and pyroptosis. A comparison of the main features within the three programmed cell death (PCD) pathways.

	Apoptosis	Necroptosis	Pyroptosis
Lytic cell death	No	Yes	Yes
Pore-forming protein	BAX/BAK	MLKL	GSDM proteins
Caspase activation	Yes	No	Yes
Inflammatory response	No	Yes	Yes
Morphological features	Cell shrinkage Nuclear condensation Membrane blebbing	Cell swelling Nuclear condensation Membrane rupture	Cell swelling Membrane rupture

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
