# Peer review of "No Time to Die: How Cytomegaloviruses Suppress Apoptosis, Necroptosis, and Pyroptosis"

_viruses, 2024, doi:10.3390/v16081272_

Round 1

Reviewer 1 Report

Comments and Suggestions for Authors

General Comments

 This review summarizes current knowledge of how cytomegaloviruses (mostly HCMV and MCMV) activate and also block pathways that lead to programmed cell death (PCD). The three PCD pathways are briefly reviewed and then mechanisms of viral antagonism are discussed. The review is well organized, and, with a few minor exceptions noted below, the writing is concise and clear. English language usage is excellent.

These are complex pathways and for the most part the authors do a good job of describing them verbally. For each PCD the authors begin by describing the pathway independent of viral antagonism. It would be helpful to refer the reader to the figures at this stage, rather than waiting to do so until subsequent sections in which viral antagonistic mechanisms are discussed.

The figures are a good start but could be more complete. There are several instances, detailed below, where factors that are discussed in the text are not found in the pathway figures; it would be helpful to include all pathways and factors in the figures. It might also be helpful to distinguish between viral factors from the two viruses by making the HCMV and MCMV antagonists different shades of the same color. 

There are several instances in which the same protein is known by two or even three names, and in some cases both names are shown in the figures while the text sometimes switches between different names. For example, UL37x1/vMIA/m38.5, m41.1/vIBO, ZBP1/DAI/DLM1, etc. This creates unnecessary confusion. It is appropriate to indicate alternative names when a factor is first mentioned in the text, but thereafter I suggest picking one name and using it consistently in both the text and the figures (don't show multiple names in the figures). When talking about gene deletion mutants it may be easier to say "deletion of the gene encoding protein X" rather than getting bogged down with explaining the name of the gene that encodes protein X.  

Specific Comments  

  1. Lines 59-60. Is "cell cycle" the right term here? Perhaps "dependent on the viability of infected cells"?
  2. Lines 101-102. It would be helpful here to clarify that these are cell surface receptors that engage extracellular ligands.
  3. Figure 1. Missing TRAIL, UL141, m166, FLICE, cFLIP, IE2 activation of cFLIP, and action of beta 2.7 RNA.
  4. Line 180. "MAVS" has not been described or defined.
  5. Line 193-195. This sentence is unclear. Why would deletion of genes encoding two viral proteins that antagonize PCD (m38.5 and m41.1) prevent PCD?
    1. HtrA2/Omi is mentioned on line 127 along with SMAC/Diablo, but there is no further discussion or explanation of what these proteins do, and they are not shown in Figure 1.
    2. Please be more specific regarding "regulate innate inflammation": enhance? impair?
  6. Line 211. Please define the acronym MLKL.
  7. Line 215. The acronym ZBP1 has been defined on line 202.
  8. Lines 231-241. M45 inhibition of NF-κB activation does not appear to be a mechanism of antagonizing PCD. If it is relevant to PCD then please provide additional context or explanation as to how. If not perhaps it should be left out?
  9. Figure 2.
    1. Missing TRL3 and TRL4.
    2. PAMPs are in this figure but are not defined until the section on pyroptosis (line 262).
    3. If NEMO and NF-κB are relevant to necroptosis they should be included in the figure.
  10. Lines 265-266. Please clarify what is meant by “NLRP3, AIM2, and RIG-I inflammasomes”.
  11. Line 271. Please clarify: does the autoproteolytic cleavage of procaspase-1 mentioned in the previous sentence result in activated “caspase-1”. Is that p20 or p10?
  12. Line 280-281. Please clarify the meaning of “mature cytokines”. What is the biological consequence of releasing “mature cytokines and other cellular components”?
  13. Line 284. Should IFI16 have been mentioned earlier, e.g., line 265?
  14. Line 285. What PAMPs do these sensors recognize?
  15. Line 289. Please specify which DNA sensors.
  16. Line 300. Please define the acronym iBMDM.
  17. Lines 303-304. Sentence is a bit unclear. M84 inhibits the proteolytic cleavage that activates caspase-1?
  18. Figure 3. Missing PAMPs, DAMPs, NLRs, ALRs, p10, p20, NINJ1, IFI16, IE2 inhibition of pro-IL-1β expression, induction of IL-1β protein degradation, ASC specks, M84 blocking of G+GSDMD cleavage, UL83.
  19. Line 327. How do CMVs “exploit PCD”?
  20. Line 351. It is unclear what is meant by “deflection”.
  21. Line 365-369. Perhaps a figure would be helpful here.

Reviewer 2 Report

Comments and Suggestions for Authors

Deng et al present a review focused on providing perspective to cytomegalovirus-encoded suppressors of cell death signaling pathways. The review is very timely as this group has recently published a viral inhibitor of inflammasome-mediated pyroptosis.

For the most part, the text is well organized; however, the meaning of many phrases need to be modified so as to convey accurate information.

Authors should comb through the entire manuscript to sort out these problems because they will leave readers a bit confused. 

As a matter of examples, here are some problematic statements and recommended edits:

Abstract, ln 8: “their replication depends entirely on the metabolism of the host cell”.  While viral replication depends on many cellular functions, it is not “entirely” dependent, so this modifier should be removed. 

Abstract, ln 9: The sentence following the one above should not start with “Therefore” given that there is no logical connection between “Viruses are obligte intracellular pathogens…” and “Therefore, the induction of cellular suicide, known as programmed cell death (PCD), can hinder viral replication and act as a first line of defense against viral pathogens”. Both thoughts should be stated independently, as “Induction of cellular suicide, known as programmed cell death (PCD), has the potential to hinder viral replication if induced prior to replication and act as a first line of defense against viral pathogens. Cell death has the potential to increase the release of viral progeny if induced later, after replication is under way”.

Abstract, ln 11: Investigators working on other cell death pathways such as ferroptosis might take exception to the statement, “Apoptosis, necroptosis, and pyroptosis are the three most important PCD modalities.” The modifier “most” can be dropped without losing meaning.

Section 1, ln 23: “Cytomegaloviruses (CMVs) are large, enveloped, double-stranded DNA viruses that infect mammals, including humans. They are members of the Betaherpesvirinae and have a narrow host range.” Would be better worded to say “Cytomegaloviruses (CMVs) are large, enveloped, double-stranded DNA viruses that infect mammals, including humans, in a species-specific manner. All CMVs are related members of the betaherpesvirinae that have diverged as they coevolved with their hosts.”

Section 1, ln 35: Authors should reword “Although the rhesus macaque CMV and the guinea pig CMV models are especially well suited to study congenital HCMV infection, these models present major disadvantages (e.g., availability of reagents and genetically modified lines, high associated costs) when compared to murine CMV (MCMV).”  Rather than point to “major disadvantages” that many investigators would disagree on, something along the lines of “Murine CMV has emerged as an animal model where the interplay of viral modulators and host defense pathways can be fully elaborated through evaluation of genetic mutants.”

Section 1, ln 53: “However, innate and adaptive immune responses are unable to clear CMV from an infected organism as the virus establishes a life-long latent infection in specific cells.” Would be more accurate as “Like all herpesviruses, CMVs remain latent within their respective hosts for life, being controlled but without being cleared by the antiviral immune response.”

Section 2, ln 82: “Because the content of the dying cell is contained within apoptotic bodies that are cleared by phagocytic cells, apoptosis is known to be a relatively inflammatory-silent form of cell death [13].” Would be more accurate if worded, “Apoptotic cells are rapidly cleared by phagocytic cells in tissues resulting in an inflammatory-silent form of cell death [13].” Furthermore, authors should appreciate that almost all studies of necroptosis and pyroptosis have been pursued in cultured cells where clearance by phagocytic cells does not occur. Thus, comparisons are difficult given that apoptosis results in secondary cell leakage in cell culture.

Section 2, ln 88: “Although these pathways are often described as distinct signaling cascades, an increasing number of studies show that there is a substantial cross-talk between these PCD forms. A mixed form of PCD, known as PANoptosis, can be particularly important in the presence of pathogen-encoded inhibitors of a specific PCD pathway (reviewed in [17,18]).” Would be much more accurately stated, “Although these pathways are often described as distinct signaling cascades, an increasing number of studies reveal a substantial level of cross-talk between PCD forms leading to alternative outcomes, such as occurs when caspase inhibition converts extrinsic apoptosis to necroptosis (ref: Upton and Chan, Mol Cell, 2014; Wallach et al., Science 2016).  A complex form of PCD, termed PANoptosis, may be triggered via Z-nucleic acid binding protein (ZBP)1 when viruses fail to suppress cell death signaling such as reported to occur following influenza A virus infection of certain cell types (reviewed in [17,18]).”

Section 2, ln 116: “The intrinsic apoptosis pathway is also known as the mitochondrial pathway and is activated when intracellular stressors interfere with the balance of the B-cell lymphoma 2 (BCL-2) family of proteins.” Would sound cleaner as “Intrinsic apoptosis (also known as the mitochondrial pathway) becomes activated when intracellular stress interferes with the balance of the B-cell lymphoma 2 (BCL-2) family of pro- and anti-apoptotic proteins.”

Section 2, ln 241: Work that has shown HSV1 and HSV2 ICP6 encodes a RHIM-mediated mechanism of suppressing necroptosis similar to MCMV M45 should be mentioned before describing the situation with HCMV in the paragraph that follows.

The section on pyroptosis is very well assembled.  The review should mention caspase-11 and relate whether the M84 gene product inhibits pyroptosis mediated by caspase-11 in addition to all the information on pyrin and the caspase-1-dependent signaling pathway.

Minor corrections:

In Fig 1, vIBO is misspelled.

Comments on the Quality of English Language

Carefully edit the language for unnecessary modifiers above and beyond those identified above.

Round 2

Reviewer 1 Report

Comments and Suggestions for Authors

The authors have adequately addressed my concerns.